# Beyond pixel-wise supervision for segmentation:
# A few global shape descriptors might be surprisingly good!

**Hoel Kervadec**[1,2]                                                        HOEL@KERVADEC.SCIENCE
[1] *ÉTS Montréal*
[2] *CRCHUM Montréal*

**Houda Bahig**[2]                                        HOUDA.BAHIG.CHUM@SSSS.GOUV.QC.CA
**Laurent Letourneau-Guillon**[2]           LAURENT.LETOURNEAU-GUILLON.1@UMONTREAL.CA
**Jose Dolz**[1,2]                                                    JOSE.DOLZ@ETSMTL.CA
**Ismail Ben Ayed**[1,2]                                         ISMAIL.BENAYED@ETSMTL.CA

**Editors:** Under Review for MIDL 2021

## Abstract

Standard losses for training deep segmentation networks could be seen as individual classifications of pixels, instead of supervising the global shape of the predicted segmentations. While effective, they require exact knowledge of the label of each pixel in an image.

This study investigates how effective global geometric shape descriptors could be, when used on their own as segmentation losses for training deep networks. Not only interesting theoretically, there exist deeper motivations to posing segmentation problems as a reconstruction of shape descriptors: First, annotations to obtain approximations of low-order shape moments could be much less cumbersome than their full-mask counterparts, and anatomical priors could be readily encoded into invariant shape descriptions, which might alleviate the annotation burden. Also, some shape descriptors could be readily used to "encode" biomarkers, leading to better interpretability. Finally, and most importantly, we hypothesize that, given a task, certain shape descriptions might be invariant across image acquisition protocols/modalities and subject populations, which might open interesting research avenues for generalization in medical image segmentation.

We introduce and formulate a few shape descriptors in the context of deep segmentation, and evaluate their potential as stand-alone losses on two different, challenging tasks. Inspired by recent works in constrained optimization for deep networks, we propose a way to use those descriptors to supervise segmentation, without any pixel-level label. Very surprisingly, as little as 4 descriptors values per class can approach the performance of a segmentation mask with 65k individual discrete labels. We also found that shape descriptors can be a valid way to encode anatomical priors about the task, enabling to leverage expert knowledge without requiring additional annotations. Our implementation is publicly available and can be easily extended: https://github.com/hkervadec/shape_descriptors.

**Keywords:** Semantic segmentation, constraints, weak supervision, shape moments.

## 1. Introduction

In the recent years, image semantic segmentation has received considerable attention in the medical imaging research community, and almost all contemporary methods rely on deep learning and fully convolutional neural networks (Ronneberger et al., 2015; Litjens et al., 2017). While network architectures have been extensively studied (Ronneberger et al., 2015; Milletari et al., 2016; Chen et al., 2017), the loss functions used to train them

have received relatively less attention, and most of the existing methods rely on variants of either the cross-entropy (Ronneberger et al., 2015) or dice loss (Sudre et al., 2017; Milletari et al., 2016). Ultimately, all those losses are actually performing "pixel-wise classification", and do not account for the image spatial domain—for instance, standard implementations in popular frameworks completely discard the image dimension[1]. Other methods that take into account the distances to the segmentation boundary (Kervadec et al., 2019a), or weakly supervised methods that have access to only partial/uncertain annotations (Qu et al., 2019; Rajchl et al., 2016; Papandreou et al., 2015; Bearman et al., 2016; Lin et al., 2016)—in a weakly supervised segmentation setting—eventually supervise a subset of pixels individually. Informally, we could say that the existing segmentation methods are "micro-managing" pixels, taking each as a separate classification problem, instead of supervising the global shape information of segmentation prediction.

The traditional computer vision literature abounds of global mathematical descriptions that characterize the shapes of objects (Nayak and Stojmenovic, 2008), for instance, shape moments, length, total variation, Fourier transforms, etc. It has also been showed that descriptions based on a few geometric shape moments could be enough to re-construct complex shapes (Milanfar et al., 2000), via solving an inverse problem. Furthermore, such geometric shape moments could be made invariant with respect to geometric transformation (e.g., rotation, translation and scaling) by pure mathematical manipulations, which is convenient for segmentation (Klodt and Cremers, 2011). This includes the well-known Hu's invariant moments (Hu, 1962). While less popular today in computer vision than they used to be, those remain powerful regularization and shape-description tools for segmentation methods. So powerful, perhaps, that they could be fully used to characterize the objects that we want to segment, while providing intrinsic invariance; in short, supervising the overall shape prediction of a segmentation networks, not through individual pixels but rather global shape descriptions.

This paper studies how effective global geometric shape descriptors can be, when used on their own as segmentation losses for training deep neural networks. Not only interesting theoretically, there exist deeper motivations to posing segmentation problems as a reconstruction of shape descriptors: First, annotations to obtain approximations of low-order shape moments could be much less cumbersome than their full-mask counterparts (e.g., from a few mouse clicks by the user). Furthermore, anatomical priors can also be readily translated into shape descriptions, which is not feasible when using dense label masks. This might alleviate the annotation burden for training deep segmentation networks. Also, some shape descriptors could be readily used to "encode" biomarkers, leading to better interpretability. Finally, and most importantly, we hypothesize that, given a task, certain shape descriptions might be invariant across image acquisition protocols/modalities and subject populations, which might open interesting research avenues for generalization in segmentation.

Our contributions can be summarized as follow:

- we re-introduce and reformulate different shape descriptors, in the context of deep semantic segmentation;

---

1. https://pytorch.org/docs/stable/generated/torch.nn.CrossEntropyLoss.html

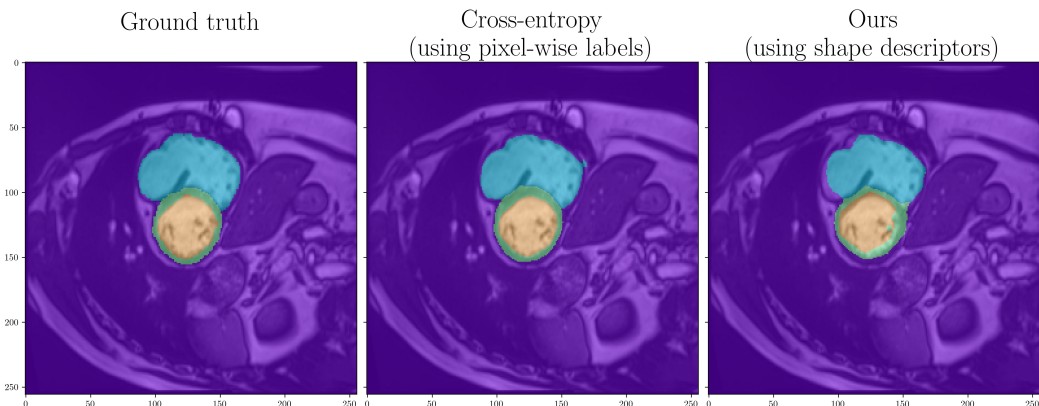

(a) A visual comparison of the different supervision methods on the ACDC dataset.

| Pixel | Label |
|:---:|:---:|
| 0 | RV |
| 1 | Background |
| 2 | LV |
| ⋮ | |
| 65536 | Background |

| Shape descriptor | Class | | |
|:---:|:---:|:---:|:---:|
| (in pixels) | RV | Myo | LV |
| Object volume $\mathfrak{V}$ | 3100 | 800 | 1600 |
| Centroid location $\mathfrak{C}$ | (125, 80) | (125, 125) | |
| Avg. dist. to centroid $\mathfrak{D}$ | (20, 15) | (15, 20) | (10, 10) |
| Object length $\mathfrak{L}$ | 750 | 1000 | 500 |

(b)     Pixel-wise  labels (**65k** discrete values)

(c) Shape descriptors (**16** continuous values)

Figure 1: RV, Myo and LV stands for "right-ventricle", "myocardium" and "left-ventricle", respectively. The shape descriptors are detailed in Subsection 2.2.

- inspired by recent works in inequality constraints, we propose a way to use those descriptors to supervise deep neural networks;

- as such, we benchmark a combination of those descriptors and show that–surprisingly– using only a few shape descriptors can go a long way, even in more complex settings (Figure 1). In fact, we found that as little as 4 descriptors values per class could approach the performance of a segmentation mask with 65k individual discrete labels.;

- we discuss future research directions that could benefit from those surprising findings.

## 2. Formulation

### 2.1. Notation and background

Let $\Omega \subset \mathbb{R}^2$ denotes the image spatial domain[2] and $I : \Omega \to \mathbb{R}$ an input image, with $G : \Omega \to \hat{\Delta}^K$ its associated ground truth. Here, $\Delta^K$ refers to the $K$-simplex, and $\hat{\Delta}^K$ to its

---

2. For readability and simplicity, we detail here only the case of 2D-images, but the method could be extended to $N$ dimensions in a straightforward way.

vertices, i.e., a one-hot encoding for $K$ classes. Our goal is to train a network $\mathcal{N}_{\boldsymbol{\theta}} : I \mapsto s_{\boldsymbol{\theta}}$, with parameters $\boldsymbol{\theta}$ predicting a dense probability map $s_{\boldsymbol{\theta}} : \Omega \to \Delta^K$; $s_{\boldsymbol{\theta}}^{(i,k)}$ denotes the predicted softmax probability for class $k \in \{0, ..., K\}$ at pixel $i \in \Omega$. For each pixel $i$, its coordinates in the 2D space are represented by the tuple $\left( x_{(i)}, y_{(i)} \right) \in \mathbb{R}^2$.

Shape and central moments have been widely studied in traditionnal computer vision (Nayak and Stojmenovic, 2008; Milanfar et al., 2000), where they can be used to characterize a shape. Each moment is parametrized by its orders $p, q \in \mathbb{N}$, and each order represents a different characteristic of the shape.

**Shape moment** Shape moments can be defined, in their general form, as functions of a deep-network softmax predictions for a given class $k$ as follows:

$$\mu_{p,q}^{(k)}(s_{\boldsymbol{\theta}}) := \sum_{i \in \Omega} s_{\boldsymbol{\theta}}^{(i,k)} x_{(i)}^p y_{(i)}^q,$$

where $p, q \in \mathbb{N}$ are the moment orders.

**Central moment** The central moment is closely related to the shape moment, with the difference being that coordinates $x_{(i)}$ and $y_{(i)}$ are shifted by their respective centroids, for translation invariance (more details in the next section). It is given by:

$$\bar{\mu}_{p,q}^{(k)} := \sum_{i \in \Omega} s_{\boldsymbol{\theta}}^{(i,k)} \left( x_{(i)} - \frac{\mu_{1,0}^{(k)}}{\mu_{0,0}^{(k)}} \right)^p \left( y_{(i)} - \frac{\mu_{0,1}^{(k)}}{\mu_{0,0}^{(k)}} \right)^q.$$

**Image Laplacian** The Laplacian of an image is defined by the underlying graph structure $\mathcal{G}_{\Omega}$, which describes the connectivity between each pair of pixels. A sparse graph (i.e., each pixel is connected only to its 4 or 8 direct neighbors) is often used, and can be encoded with a sparse adjacency matrix $A_{\Omega} \in \{0, 1\}^{|\Omega| \times |\Omega|}$, where $A_{\Omega,i,j} = 1$ means that $i, j$ are neighbors, and $A_{\Omega,i,j} = 0$ means that they are not. The Laplacian can be directly constructed from $A_{\Omega}$:

$$L_{\Omega} := A_{\Omega} - \mathrm{diag}(\mathbf{1}^{\top} A_{\Omega}),$$

where $\mathrm{diag} : \mathbb{R}^a \to \mathbb{R}^{a \times a}$ builds a diagonal matrix and $\mathbf{1}^{\top} A_{\Omega} \in \mathbb{R}^{|\Omega|}$ encodes the number of neighbors for each pixel $i$. For a 8-neighbors connectivity, this number will be the same for all pixels, except the image edges which will have less.

Notice that $L_{\Omega}$ depends only on the image spatial domain, but not the image values. As such, it can be efficiently pre-computed and cached for all the samples in a dataset. There exists some edge-sensitive variants, which define $A_{\Omega}$ so that it accounts for pixel similarities (ex., intensity differences), but this is beyond the scope of this study.

## 2.2. Shape descriptors

With those different standard building blocks, it is possible to define shape descriptors, measuring actual properties of the object, rather than listing a list of pixels that should belong to it. All following descriptors hold for some input image $I$ and some class $k$:

**Volume** The volume of the predicted segmentation is simply a summation of the predicted probabilities—which is a special case of shape moments. As such:

$$\mathfrak{V}^{(k)}(s_{\boldsymbol{\theta}}) := \mu_{0,0}^{(k)}(s_{\boldsymbol{\theta}}).$$

**Centroid** The centroid of a class can be computed by dividing the first shape moment by the volume. We can see it as the average of the pixel-coordinates for class $k$:

$$\mathfrak{C}^{(k)}(s_{\boldsymbol{\theta}}) := \left( \frac{\mu_{1,0}^{(k)}(s_{\boldsymbol{\theta}})}{\mu_{0,0}^{(k)}(s_{\boldsymbol{\theta}})}, \frac{\mu_{0,1}^{(k)}(s_{\boldsymbol{\theta}})}{\mu_{0,0}^{(k)}(s_{\boldsymbol{\theta}})} \right).$$

**Average distance to the centroid** It measures how far the object should spread around its centroid, *on average.* It is the standard deviation of pixel-coordinates for class $k$:

$$\mathfrak{D}^{(k)}(s_{\boldsymbol{\theta}}) := \left( \sqrt[2]{\frac{\bar{\mu}_{2,0}^{(k)}(s_{\boldsymbol{\theta}})}{\mu_{0,0}^{(k)}(s_{\boldsymbol{\theta}})}}, \sqrt[2]{\frac{\bar{\mu}_{0,2}^{(k)}(s_{\boldsymbol{\theta}})}{\mu_{0,0}^{(k)}(s_{\boldsymbol{\theta}})}} \right).$$

**Length** The length of a segmentation, or rather, the length of its boundary, can be efficiently computed by re-using the pre-computed image Laplacian. To summarily describe it, each difference of classification between two neighbors will be counted as 1, while neighbors with the same predicted class will count as 0; which is a standard Potts model. It is trivial to relax this definition to plug the predicted (continuous) probabilities:

$$\mathfrak{L}^{(k)}(s_{\boldsymbol{\theta}}) := \sum_{i,j \in \mathcal{G}_{\Omega}} |s_{\boldsymbol{\theta}}^{(i,k)} - s_{\boldsymbol{\theta}}^{(j,k)}| L_{\Omega,i,j}.$$

**Ratio of descriptors** In the multi-class setting, some relationships between different classes might be known in advance, using anatomical priors, for instance. While exact values are not necessarily required, inequalities could provide useful information. As such, we can define an additional descriptors for pairs of classes $k$ and $l$, for a specific descriptor $\mathfrak{f} \in \{\mathfrak{V}, \mathfrak{C}, \mathfrak{D}, \mathfrak{L}\}$:

$$\mathfrak{R}_{\mathfrak{f}}^{(k,l)}(s_{\boldsymbol{\theta}}) := \frac{\mathfrak{f}^{(k)}(s_{\boldsymbol{\theta}})}{\mathfrak{f}^{(l)}(s_{\boldsymbol{\theta}})}.$$

### 2.3. Supervision with constraints

Instead of optimizing a pixelwise loss, we design loss functions, which penalize the deviations between the global shape descriptors computed from the predicted segmentation and those corresponding to the ground truth, e.g., $\mathfrak{C}^{(k)}(s_{\boldsymbol{\theta}}) = \mathfrak{C}^{(k)}(G)$. This could be formulated as a hard equality-constrained optimization problem. Here, we propose to relax the constraint to add a lower and upper bound centered around the ground truth value (this may mimic imprecise information about shape descriptors when these are derived, for instance, from anatomical prior knowledge and not from ground truth):

$$\arg\min_{\boldsymbol{\theta}} \quad \mathcal{L}_{\boldsymbol{\theta}} \tag{1}$$
$$\text{subject to} \quad 0.9\tau_{\mathfrak{f}}^{(k)} \leq \mathfrak{f}^{(k)}(s_{\boldsymbol{\theta}}) \leq 1.1\tau_{\mathfrak{f}}^{(k)} \qquad \forall k, \ \forall \mathfrak{f} \in \{\mathfrak{V}, \mathfrak{C}, \mathfrak{D}, \mathfrak{L}\}$$
$$a \leq \mathfrak{R}_{\mathfrak{f}}^{(k,l)} \leq b \qquad \text{for some } \mathfrak{f}, a, b, k, l,$$

where $\tau_{\mathfrak{f}}^{(k)} = \mathfrak{f}^{(k)}(G)$.

In the context of deep neural networks, standard constrained optimization techniques (such as Lagrangian or interior-point methods) are not directly applicable for tractability reasons. The inequality constraints can be tackled directly as a loss function using a *log-barrier-extension* penalty $\widetilde{\psi}_t(\cdot)$ (details can be found in Section A), controlled by a parameter $t$ that is increased over time to make the bounds tighter and tighter. Such log-barrier penalties were introduced recently in (Kervadec et al., 2019c) in the general optimization context for constrained deep networks. As, for the sake of the study, we want to completely forego pixel-wise supervision, we set $\mathcal{L}_{\boldsymbol{\theta}} := 0$. As such, our final model is:

$$\arg\min_{\boldsymbol{\theta}} \quad \sum_{\mathfrak{f}} \sum_{k} \left[ \widetilde{\psi}_t \left( 0.9\tau_{\mathfrak{f}}^{(k)} - \mathfrak{f}^{(k)}(s_{\boldsymbol{\theta}}) \right) + \widetilde{\psi}_t \left( \mathfrak{f}^{(k)}(s_{\boldsymbol{\theta}}) - 1.1\tau_{\mathfrak{f}}^{(k)} \right) \right]. \tag{2}$$

Bounds for $\mathfrak{R}_{\mathfrak{f}}^{(k,l)}$ can be included in the same fashion, if available and relevant—depending on the task at hand. The bound values for $\mathfrak{R}$ do not rely on $G$, but rather on expert knowledge about the task. We will give some examples in the next section.

## 3. Experiments

### 3.1. Datasets

**Heart segmentation on cine-MRI** The main dataset that we use in our experiments is the publicly available 2017 ACDC Challenge (Bernard et al., 2018), which contains 4 classes to segment: left and right ventricles, myocardium, and background. The dataset consists of 100 cine magnetic resonance (MR) exams covering well defined pathologies: dilated cardiomyopathy, hypertrophic cardiomyopathy, myocardial infarction with altered left ventricular ejection fraction and abnormal right ventricle. It also included normal subjects. We chose this dataset because it is a good benchmark for shape descriptors. Not only a multi-class setting, the myocardium and left-ventricle share a common centroid, and the myocardium completely surround the left-ventricle—which is more challenging to describe. We constraint $\mathfrak{S}, \mathfrak{C}, \mathfrak{D}, \mathfrak{L}$, and $\tau_{\mathfrak{C}}^{(\text{MYO})} = \tau_{\mathfrak{C}}^{(\text{LV})}$. Moreover, the relationship between myocardium and left-ventricle can be formulated with the following bounds: $2 \leq \mathfrak{R}_{\mathfrak{L}}^{(\text{MYO,LV})} \leq 3$. We retained 70 exams for training, 10 for validation and 20 for testing.

**Prostate segmentation on MR-T2** The second dataset that we use is the PROMISE12 challenge (Litjens et al., 2014). It contains the transversal T2-weighted MR images of 50 patients acquired at different centers, with multiple MRI vendors and different scanning protocols. The images include patients with benign diseases, as well as with prostate cancer. We employed 35 patients for training, 5 for validation, and 10 for testing. The difficulty of this dataset lies in the low-contrast, and very variable shape of the prostate. We supervise $\mathfrak{S}, \mathfrak{C}, \mathfrak{D}, \mathfrak{L}$.

### 3.2. Implementation details

We use the ENet architecture (Paszke et al., 2016) for experiences on ACDC, and a modified fully residual UNet for the experience on PROMISE12—the prostate is a harder task that requires a more powerful network, and it also enables us to validate the supervision method on different architectures. We perform blurring, shifting, and scaling as online data

augmentations, and we use the same network initialization for all settings, with the same scheduler and hyper-parameters (Adam scheduler (Kingma and Ba, 2014), with learning rate of 5e-4 and $\beta = (0.9, 0.99)$). The shape descriptors are computed from the annotated mask, and we use a relaxed value by $\pm 10\%$ as bounds. Most of the implementation was done in the PyTorch framework, and experiments were run on a Nvidia Titan RTX. All descriptors can be efficiently vectorized, resulting in minimal slowdown during training (less than 10% compared to a cross-entropy loss). The computation of the Laplacian $\mathcal{L}_\Omega$ is done once per image shape (usually a single one per dataset after pre-processing), and cached using standard Python utilities (`lru_cache` from `functools`). Our code is publicly available at https://github.com/hkervadec/shape_descriptors, and can easily be extended to other shape descriptors.

## 4. Results

Surprisingly, using only a few shape descriptors in place of dense pixel-wise supervision is capable to segment the objects of interest, as we can see in Figure 2. On ACDC, what remains the most difficult to learn is the hierarchy between the left-ventricle and its surrounding myocardium, and some noisy myocardium pixels can sometime remain inside the predicted left-ventricle. Nonetheless, we can consider that the network has properly learned the overall structure of the heart. On Promise12, the task is difficult even for cross-entropy with full annotations. Despite the more powerful network used, the low-contrast can still trick both methods. Nevertheless, supervision with shape descriptors is capable to get a rough location and shape of the prostate, which is much more than we initially expected. Actual testing DSC values can be found in Table 1, and the plots of training and validation metrics over time can be found in Appendix B.

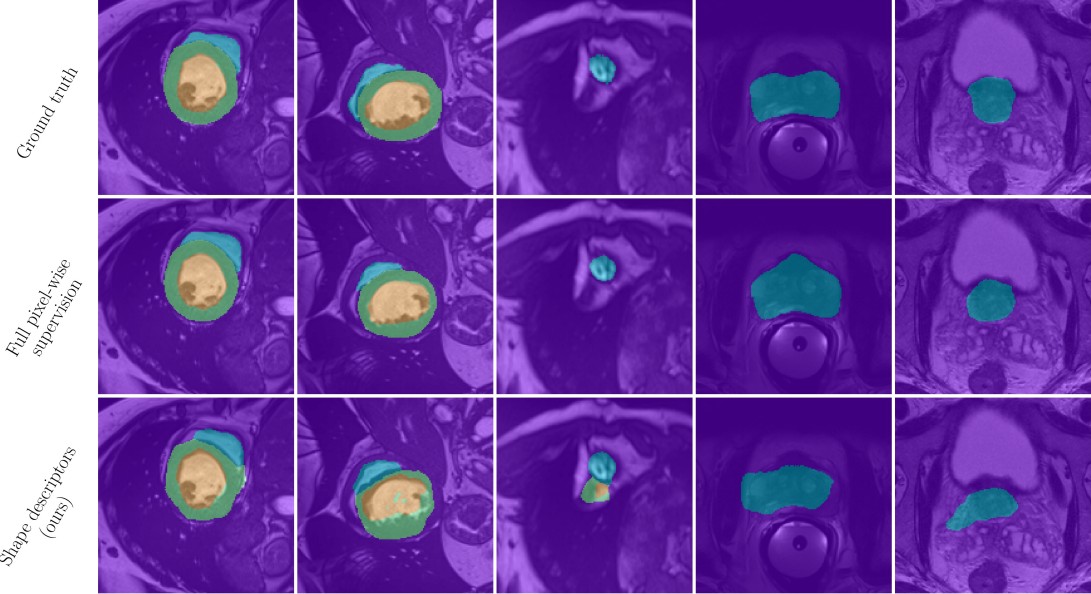

Figure 2: Visual comparison for both ACDC and Promise12 on the testing set, including some failure cases.

Table 1: Average and standard deviation of DSC on the testing set, for both datasets.

| | ACDC | | | | Promise12 |
|---|---|---|---|---|---|
| Method | RV | Myo | LV | Overall | Prostate |
| Cross-entropy (pixel-wise) | 0.879 (0.066) | 0.829 (0.074) | 0.919 (0.059) | 0.876 (0.076) | 0.871 (0.047) |
| Ours (shape descriptors) | 0.825 (0.107) | 0.660 (0.114) | 0.819 (0.086) | 0.768 (0.128) | 0.651 (0.098) |

## 5. Discussion and conclusion

We have showed that simple and light shape descriptors can be effective supervision tools for semantic segmentation, allowing us to avoid completely pixel-wise supervision; proving how powerful shape descriptors can be. In a multi-class setting, the neural network is able to learn the inherent relationship between classes and the anatomical structure of the heart.

While not needed on the two datasets that we benchmarked on, it is very easy to compute the orientation and elongation of an object (Nayak and Stojmenovic, 2008), which would be very useful for certain tasks (for instance, esophagus segmentation). Spatial relationship between classes, that would be translation invariant, could be very beneficial in some settings, such as the co-segmentation of esophagus and trachea—both long objects, next to each others.

We found empirically that using only shape descriptors without online data augmentation was more sensitive to network initialization than its pixel-wise counterpart. It is entirely plausible that the random networks' initializations, designed and tuned with cross-entropy in mind (Sutskever et al., 2013), are not optimal for shape descriptors. As such, future works could investigate other network initialization strategies.

One main limitation of the method is its inability to be sub-patched and processed in different batches (Any loss requiring a sum over an area bigger than the current patch shares this limitation, including the very popular Dice loss and its derivatives.) Recently, for a similar ill-suited problem (enforcing a prior of the distribution of the classes, over the whole training set), (Zhou et al., 2019) showed that a primal-dual approach can be a promising avenue.

We believe that we barely scratched the surface for the potential of invariant shape descriptors: shape and central moments orders can go much higher than two. Depending on the task, some invariant and higher-order descriptors could be common to all the samples and would not require additional annotations, but rather exploit existing anatomical knowledge. This might open interesting avenues for generalization across subject populations and acquisition protocols. Also, using the Discrete Fourier Transform to characterize the objects in the frequency domain could provide an interesting avenue for future works.

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

## Appendix A. Extended log-barrier

The extended log-barrier was introduced in (Kervadec et al., 2019c), as standard Lagrangian or interior-point methods are not directly applicable to deep learnign settings. If we take a simple constrained optimization setting:

$$\arg\min_{\boldsymbol{\theta}} \quad \mathcal{L}_{\boldsymbol{\theta}}$$
$$\text{subject to} \quad z \leq 0,$$

then its extended log-barrier equivalent is:

$$\arg\min_{\boldsymbol{\theta}} \quad \mathcal{L}_{\boldsymbol{\theta}} + \widetilde{\psi}_t(z)$$

$$\widetilde{\psi}_t(z) = \begin{cases} -\frac{1}{t}\log(-z) & \text{if } z \leq -\frac{1}{t^2} \\ tz - \frac{1}{t}\log(\frac{1}{t^2}) + \frac{1}{t} & \text{otherwise,} \end{cases}$$

where $t$ is the slope parameter of the log-barrier that is increased over time, eventually "closing" the barrier when $t \to \infty$. This is illustrated in Figure 3

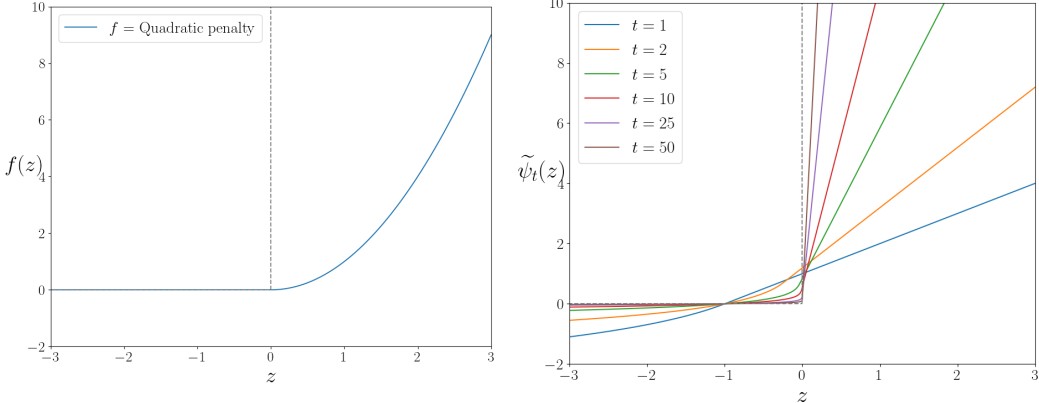

Figure 3: Illustration of the extended log-barrier (Kervadec et al., 2019c), for increasing $t$ values, compared to a fixed quadratic penalty (Kervadec et al., 2019b).

The advantage of the log-barrier are two-fold:

- it allows to gradually increase the tightness of the constraints that we want to satisfy;

- once satisfied, it gently pushes back the constrained function toward the feasible set, preventing it to go out of bounds.

## Appendix B. Training curves

The training curves shows that the trainig is fairly stable over time, though in the case of PROMISE12 it takes a few epochs for the network to start producing meaningful predictions. This is related, we think, to the random initialization procedure used in standard deep learning settings, which might not be the most optimal method when using different forms of supervision.

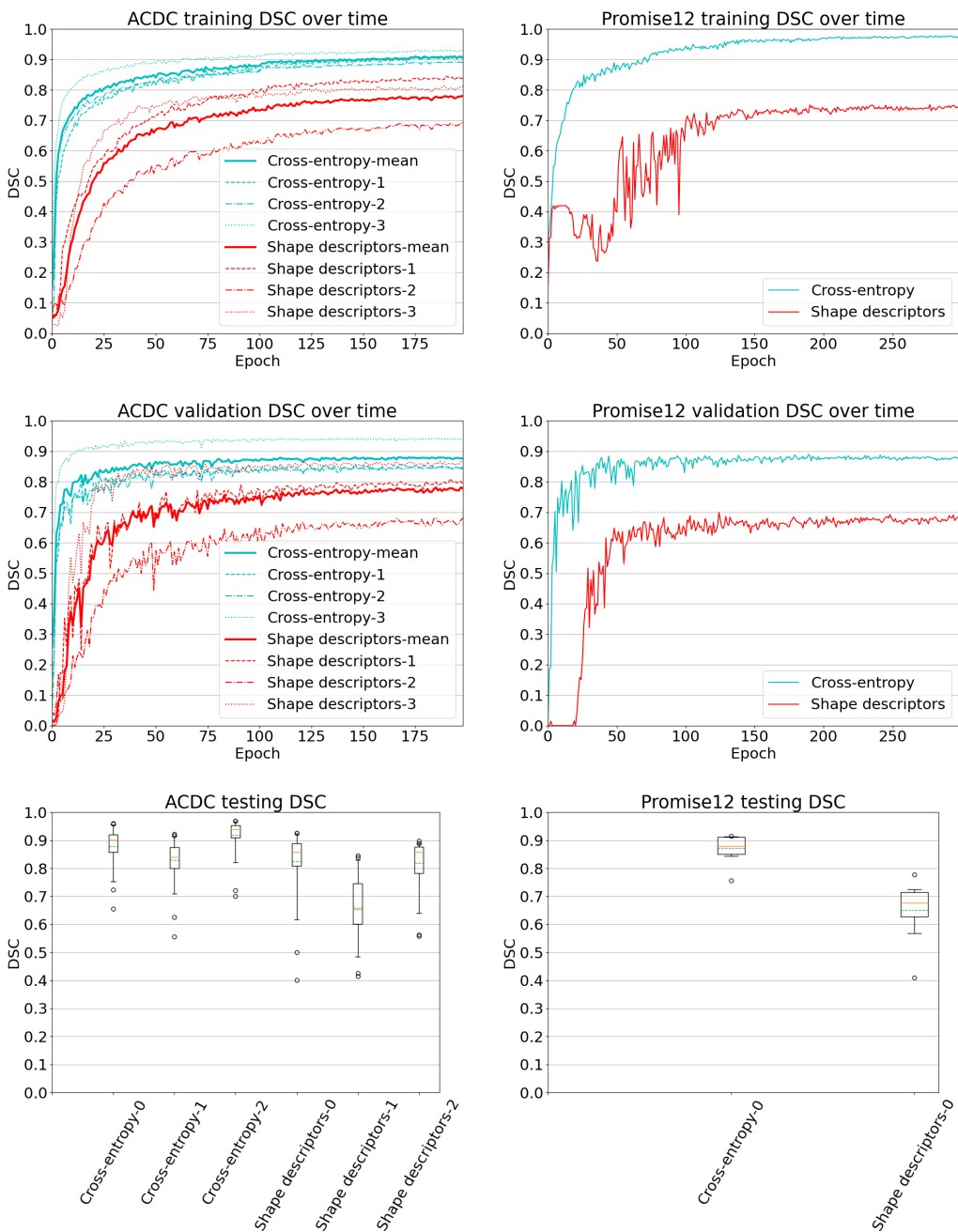

Figure 4: Comparison of the training and validation dice, and the distribution of (patient-wise) testing dice for each individual class, for both datasets. In the case of ACDC, we plot one curve per class (RV=1, Mʏᴏ=2, LV=3), as well as an average of the three.

