# OpenReview forum: "Beyond pixel-wise supervision: semantic segmentation with higher-order shape descriptors"
_MIDL.io/2021/Conference — MIDL 2021_

### Official Review · AnonReviewer3 · 2021-03-06

**Confidence:** 3
**Preliminary Rating:** 4
**Recommendation:** Best Paper Award, Oral
**Final Rating:** 4

**Summary:**

This paper investigates using Global descriptors as loss functions to optimize segmentation networks instead of the traditional approach of treating the problem as pixel classification. The authors demonstrate how they can be implemented as a Deep Learning loss, and achieve decent results considering the ease of manual annotation for the use of global descriptors in comparison to the traditional “pixel classification” approach.


**Strengths:**

The paper is well written and concise, while being mathematically rigorous in describing all the descriptors applied by the authors. The idea is very interesting and opens the door for future works to explore manual annotations with shape descriptors instead of a detailed pixel/voxel wise annotation, besides many possible variations using the descriptors described on the manuscript. Additionally, the proposed loss has potential to bring specialist input beyond annotation of what location  is and isn’t the target structure in the image.

Special attention was given to demonstrations of how all shape descriptors can be vectorized and applied in practice, therefore, able to be used for deep learning. The included code is of high quality and very useful for reproducibility.

Figures are well made and explanations of how the method works are clear, while also presenting detailed information. The appendix sections present more detail with well made figures, and some necessary additional implementation and training explanations.


**Weaknesses:**

The only point i would highlight as a weakness is that more validation is necessary, especially analysis of each descriptor’s contribution, as in a single descriptor loss. For some intuitively this would only lead to no convergence, but even that experimental information would be useful in my opnion.

Could using descriptors as a second component to Cross Entropy improve performance? Understandably this is a methodological proposal and not a validation/application paper, but more practical experiments in the behaviour of deep architectures under this kind of supervision would enrich the contribution. I am sure the authors see the many variations that could be tested, as is discussed on the end of the paper.


**Deanonymize Review:**

no

**Detailed Comments:**

I see no minor improvements necessary to this submission.

**Final Rating Justification:**

The authors gave very complete answers to my points and others. Congratulations for your work.

**Justification Of The Preliminary Rating:**

The paper is very well written, presenting a demonstration of how shape descriptors can be adapted to be used in a semantic segmentation supervision setting, with initial steps presented in its practical usefulness. An idea that would interest the community and maybe open the doors for future work in “shape supervision”. As far as I know, there is no similar work published in other venues.


**Paper Type:**

methodological development

**Questions To Address In The Rebuttal:**

Do you have any information on which of the shape descriptors contributed more to the final results? Maybe some of them are not even necessary.

What is your opinion on extension of these shape descriptors to be used on 3D CNNs? Would it be feasible especially considering the caching approach?

Have you experimented into not using the log barriers when using the shape descriptors? Finally, what about using this loss as a second component to Cross Entropy (which would require the full annotation)?





**Special Issue:**

yes

---

> ### Author Response · Authors · 2021-03-18
> **Response to Reviewer 3**
>
> ### Weaknesses
> > The only point i would highlight as a weakness is that more validation is necessary, especially analysis of each descriptor’s contribution, as in a single descriptor loss. For some intuitively this would only lead to no convergence, but even that experimental information would be useful in my opnion.
>
> We will perform an exhaustive ablation study in a future extension, by both removing some descriptors completely, or removing descriptors for only some classes (in the multi-class setting). (For instance, `myocardium = image volume - right ventricle - left ventricle - background`, so skipping its volume information might not impact performances much.)
>
> > Could using descriptors as a second component to Cross Entropy improve performance? Understandably this is a methodological proposal and not a validation/application paper, but more practical experiments in the behaviour of deep architectures under this kind of supervision would enrich the contribution. I am sure the authors see the many variations that could be tested, as is discussed on the end of the paper.
>
> > Finally, what about using this loss as a second component to Cross Entropy (which would require the full annotation)?
>
> We can expect some benefits in this setting as well, though this is not a novel result (for instance, length/perimeter regularization has been widely used in graph-cut and total variation methods).
>
> Another, big opportunity that we see is for semi-supervised settings (as described in the top-level comment), where the descriptors could be "mined" from the annotated set, and used to regularize the unannotated set.
>
> ### Questions To Address In The Rebuttal
> > Do you have any information on which of the shape descriptors contributed more to the final results? Maybe some of them are not even necessary.
>
> We have not completed yet an exhaustive ablation study, but it seems that the shape ratio adds less value (but still some), as we already provide the shape values for both classes. We intend in the future to perform a comparison where we provide only one shape value, and keep the ratio: for instance, providing the length of the left-ventricle and withholding the length of the myocardium.
>
> > What is your opinion on extension of these shape descriptors to be used on 3D CNNs? Would it be feasible especially considering the caching approach?
>
> All descriptors can be extended fairly easily in 3D, the issue remains to fit the whole volume at once into memory to be able to compute and back-propagate on a global statistic.
>
> This is quickly problematic for 3D-CNN, but for 2D networks there are some ways. As we did in [1], 2D slices of a single 3D volume can be regrouped into the same batch, enabling the computation of 3D statistics (for instance, when computing the size, the einsum notation `bkhw->bk` becomes `zkyx->k`, where `b: batch size`, `k: number of classes`, `hw: height width` and `zyx: 3d axises`). With efficient neural networks such as ENet, the maximum batch size at training can go quite high (around `96x256x256` on a 16GB card, `140x256x256` on a 24GB card, which are becoming more and more common). Depending on the task, as noted in [2], 3d segmentation is not necessarily better than a series of 2d segmentation, so the approach is viable.
>
> > Have you experimented into not using the log barriers when using the shape descriptors?
>
> So far, no. Based on the results in [3, 4], other existing constraining methods failed completely in more complex settings so we have not tried them yet. As for minimizing the descriptor directly, it is possible to do so for the length (though it introduces a shrinking bias), but it is not possible for the other ones (or rather, it does not make sense to do so).
>
>
> ---
> [1] Constrained-CNN losses for weakly supervised segmentation, Kervadec et al, MedIA 2019
>
> [2] An exploration of 2D and 3D deep learning techniques for cardiac MR image segmentation, Baumgartner et al, MICCAI workshop 2017
>
> [3] Bounding boxes for weakly supervised segmentation: Global constraints get close to full supervision, MIDL 2020
>
> [4] Constrained Deep Networks: Lagrangian Optimization via Log-Barrier Extensions, arxiv preprint

---

### Official Review · AnonReviewer4 · 2021-03-07

**Confidence:** 4
**Preliminary Rating:** 4
**Recommendation:** Oral
**Final Rating:** 4

**Summary:**

The paper described a strategy to train deep neural networks with shape descriptors only, without needing the dense pixel-wise annotations. The authors described multiple shape descriptors: volume, centroid, the average distance to centroid, length, and the ratio of shape descriptors between classes. The proposed approach is tested on two different datasets and results are compared with training with dense pixel-wise annotations.

**Strengths:**

Key strengths of the papers are:
1) clearly laying the motivation and possible advantage of using shape descriptors in training instead of dense annotations
2) clear description of methods, choosing an intuitive baseline method for comparison, honest discussion on limitations and future directions
3) the paper is well structured and easy to follow.

**Weaknesses:**

The presented experiments in the paper highlight that a decent segmentation performance can be achieved by using shape descriptors, which can be computed from comparatively sparse annotations, hence saving manual efforts in the annotations. As such, the proposed approach is not beneficial for scenarios where dense annotations are already available. It would have been interesting to also investigate if loss based on shape descriptor provides an additional benefit when used in combination with cross-entropy loss on pixel-wise annotations. If it has an additional benefit, using shape descriptor based loss will be beneficial for the densely annotated dataset as well.

**Deanonymize Review:**

no

**Final Rating Justification:**

The original content of the paper was already good. The authors have answered a few minor points raised during the review.

**Justification Of The Preliminary Rating:**

The paper describes a novel method to train deep neural networks for medical image segmentation. The proposed shape descriptor based losses are fundamentally promising to incorporate anatomical priors and global properties of organs, and improve the generalization of automatic segmentation approaches. The authors have convincingly demonstrated the potential of their proposed approach. Though the results are not super-exciting, and more experiments could be added, but as such, the paper is a good contribution to the scientific community.

**Paper Type:**

methodological development

**Special Issue:**

no

---

> ### Author Response · Authors · 2021-03-18
> **Response to Reviewer 4**
>
> ### Weaknesses
> > The presented experiments in the paper highlight that a decent segmentation performance can be achieved by using shape descriptors, which can be computed from comparatively sparse annotations, hence saving manual efforts in the annotations. As such, the proposed approach is not beneficial for scenarios where dense annotations are already available. It would have been interesting to also investigate if loss based on shape descriptor provides an additional benefit when used in combination with cross-entropy loss on pixel-wise annotations. If it has an additional benefit, using shape descriptor based loss will be beneficial for the densely annotated dataset as well.
>
> We can expect some benefits in this setting as well, though this is not a novel result (for instance, length/perimeter regularization has been widely used in graph-cut and total variation methods).
>
> We also refer to the top-level comment, where we describe a semi-supervised setting that would combine the strength of the pixel-wise annotations and the shape descriptors, by "mining" descriptors on the annotated data and apply them to the unannotated data.

---

### Official Review · AnonReviewer1 · 2021-03-08

**Confidence:** 4
**Preliminary Rating:** 3

**Summary:**

Investigates how effective global geometric shape descriptors could be, when used on their own as segmentation losses for training deep networks.
(re)Introduce and formulate shape descriptors in the context of deep segmentation,
Evaluate their potential as stand-alone losses on two different segmentation tasks
Based on own recent work in constrained optimization for deep networks
Only 4 descriptors values per class can approach performance of standard pixel-wise trained segmentation



**Strengths:**

Github link to code: https://github.com/hkervadec/shape_descriptors.
Nice change of paradigm
Nice revisit of computer-vision shape characterisitcs (e.g moments)
Original idea based on previous methodological work from same group to formulate loss functions on shape parameters.





**Weaknesses:**

Not sure to have understood if background is a class treated like other structures, hence with computation of the shape descriptors.

Interesting idea, but results remain sub-optimal for exploitation needing detailed and precise contours (eg myocardium Dice=0.66). Not sure what the real application is.

The computation of the Laplacian LΩ is done once per image shape (usually a single one per dataset after pre-processing),: I did not follow this.

**Deanonymize Review:**

no

**Justification Of The Preliminary Rating:**

Strong background methodological publication on the shift in paradigm
Illustrations on open-data and with honest documentation of results (some far from perfect)
Generic framework that could be investigated in many applications

**Paper Type:**

validation/application paper

**Special Issue:**

no

---

> ### Author Response · Authors · 2021-03-18
> **Response to Reviewer 1**
>
> ### Weaknesses
> > Not sure to have understood if background is a class treated like other structures, hence with computation of the shape descriptors.
>
> We will rework that part in the camera ready version. The background is usually not supervised, and most descriptors would make little sense for it. However, for the prostate setting only (binary task), we found that supervising the background volume (but not its other descriptors) helped.
>
> > Interesting idea, but results remain sub-optimal for exploitation needing detailed and precise contours (eg myocardium Dice=0.66). Not sure what the real application is.
>
> This paper started as a benchmark of shape descriptors, to see 'how far' they could go. As we now know that they can be quite powerful, we are starting to see new applications that did not occur to us in the first place (see the top-level comment). We believe that with this new framing of segmentation problems, different uses will gradually appear.
>
> > The computation of the Laplacian LΩ is done once per image shape (usually a single one per dataset after pre-processing),: I did not follow this.
>
> The Laplacian of an image is actually the Laplacian of its underlying graph, which is a grid. That grid structure and size is common to all images sharing the same resolution (for instance, 256x256 after pre-processing). This makes caching possible and very efficient.
>
> There exist variants of Adjacency graph/Laplacian for images that use intensity differences to weight each edge (as in a GridCRF), which is then useful to fit the segmentation to the image content. However, they present two main limitations:
> - the hyperparameters of the intensity difference are difficult to set when images have varying levels of contrast across the dataset;
> - $L_\Omega$ is now $L_I$, which is unique to each image, preventing caching. This is not a limitation in theory, but in practice, support for sparse tensors in popular frameworks is rather poor, making it less efficient than it should be. (For Pytorch, sparse tensors have been in beta for years without significant changes, and passing sparse tensors across dataloaders is still problematic.)

---

### Official Review · AnonReviewer2 · 2021-03-08

**Confidence:** 4
**Preliminary Rating:** 3
**Recommendation:** Best Paper Award, Oral
**Final Rating:** 4

**Summary:**

This paper proposes to use global shape descriptors as loss function for training a DL model for segmentation. The efficacy of this idea is demonstrated in two challenging segmentation tasks. The effect of using low-order shape moments has the potential of reducing the workload on clinicians involved in the annotation. Furthermore it could translate in biomarker discovery, since shape plays a big role in many pathologies.

**Strengths:**

The authors show that by using ONLY 4 descriptors (16 continuous values), a level of performance comparable to that obtained with a dense annotation can be achieved. This is impressive. I particularly like the idea and the proof of how softmax probabilities translate to shape and central moments.

Not extensive experimental validation.

**Weaknesses:**

The shape descriptors utilized to train the networks were extracted from dense annotation, and while for regular and almost spherical shape, manual annotation of these shape descriptor could be as simple as requiring a few mouse clicks (it is still very difficult to be precise with volumetric data), for other structures it is way more difficult to estimate the suggested descriptors.


**Deanonymize Review:**

no

**Detailed Comments:**

The paper is very well written, but there are many grammar mistakes (e.g. been showed; capable to get) for which I would recommend careful proofreading.
Comments in the abstract about biomarker discovery, interpretability and invariance across data are very interesting. Have the authors investigated to which extend generalization across imaging modalities apply especially when major covariate shifts are present? would the approach still require forms of domain adaptation?
Looking at the results reported in Table 1, performances on myocardium and prostate segmentation are substantially lower - it would be interesting to have a figure which shows how segmentation looks like in the best and worse cases. A follow up question would involve thoughts on how segmentation could be improved, would a larger training set be required, or additional shape descriptors?
It is interesting that network initialization had more than an effect on training stability.


**Final Rating Justification:**

It has a solid foundation and as the authors said this is the beginning of a new exciting time where shape can potentially be integrated to deep learning algorithms, and with more work hopefully reach a competitive performance with that obtained using dense per-pixel approaches.

**Justification Of The Preliminary Rating:**

I really like the idea, but further experiments are recommended to demonstrate how reliable the segmentation is when it comes to using only a few shape descriptors. Without a larger number of validating experiments it is hard to really appreciate the potential of the method.

**Paper Type:**

methodological development

**Questions To Address In The Rebuttal:**

I assume that working without dense pixel annotations is the ultimate goal of the approach.
As stated above, in some cases it would be rather difficult to define GT descriptors when no dense annotation is available - investigate the affect of label noise  (e.g. imprecise centroid location, or volume estimate) in terms of dice similarity against correct dense targets.

Consider that in certain domains, an erroneous positioning of the centroid, elongation estimate or other shape descriptors may highly impact the tissue segmentation quality. A problem of this complexity can be the segmentation of the knee cartilage for which public datasets are available.
Furthermore if the segmentation is used in a dowstream tasks, let's say tissue relaxation maps, you might be estimating these biomarkers on different structures making the obtained value no more meaningful.

Ultimately, among others, Milletari et al at MICCAI 2017 introduced shape priors to CNNs, it would be nice to see comments on what are pros and cons of the two approaches.

**Special Issue:**

yes

---

> ### Author Response · Authors · 2021-03-18
> **Response to Reviewer 2 (part2)**
>
> ### Questions to address in the rebuttal
> > I assume that working without dense pixel annotations is the ultimate goal of the approach. As stated above, in some cases it would be rather difficult to define GT descriptors when no dense annotation is available - investigate the affect of label noise (e.g. imprecise centroid location, or volume estimate) in terms of dice similarity against correct dense targets.
>
> Currently, as we used a range of values to constraint the training (taking the exact computed value and adding +- 10%), the information fed to the network is already somewhat "imperfect" (but not downright noisy). We intend, in a future extension, to push the bounds more loosely, and also to add some "corruption" into the labels. We might do it by adding noise into the initial dense annotation, and see how both methods reacts to it.
>
>
> > Consider that in certain domains, an erroneous positioning of the centroid, elongation estimate or other shape descriptors may highly impact the tissue segmentation quality. A problem of this complexity can be the segmentation of the knee cartilage for which public datasets are available. Furthermore if the segmentation is used in a dowstream tasks, let's say tissue relaxation maps, you might be estimating these biomarkers on different structures making the obtained value no more meaningful.
>
> Some applications will indeed be more sensitive to label noise. We thank the reviewer for suggesting the task of knee cartilage segmentation, as it seems to be a good task to study it.
>
> > Ultimately, among others, Milletari et al at MICCAI 2017 introduced shape priors to CNNs, it would be nice to see comments on what are pros and cons of the two approaches.
>
> Due to the current space constraints (hard-limit at 8 pages), we will add an extended related works section in the appendix. But to summarize it here:
>
> [1] integrate PCA values (computed from the annotated dataset) into a dedicated "PCA-aware" CNN architecture. This was used to regularize the shape of the predicted segmentation, based on the priors computed previously. We see two limitations here: i) the custom architecture makes it incompatible with existing FCN architecture (such as U-Net) often used for segmentation (whereas descriptors are plugged to existing frameworks as an additional loss) ii) the shape priors, computed through PCA, might be harder to interpret.
>
> We will also add a discussion on [2], published at MIDL 2020, that integrated a "star-shape" prior into their training, for 2D segmentation. This was used to regularize the contour of ventricle-segmentation on 2D slices, around a used-provided centroid. The original prior (Ray et al. 2012) was optimized using dynamic programming, which isn't usable for deep neural networks. In that paper, the authors showed that it was possible to "learn" that dynamic programming part using an additional, trainable network module. However, and this is the main downside of the method, it requires user-provided centroid **even at inference**.
>
> ----
> [1] Integrating statistical prior knowledge into convolutional neural networks, Milletari et al, MICCAI 2017
>
> [2] End-to-end learning of convolutional neural net and dynamic programming for left ventricle segmentation, Nguyen et al, MIDL 2020

---

> ### Author Response · Authors · 2021-03-18
> **Response to Reviewer 2 (part1)**
>
> ### Weaknesses
> > The shape descriptors utilized to train the networks were extracted from dense annotation, and while for regular and almost spherical shape, manual annotation of these shape descriptor could be as simple as requiring a few mouse clicks (it is still very difficult to be precise with volumetric data), for other structures it is way more difficult to estimate the suggested descriptors.
>
> Indeed estimating descriptors from sparse points might be difficult for more complex shapes. However, we think that the main interest of descriptors might be their 'generalization' to other scan (we describe one such setting in the top-level comment). We hypothesize that on some tasks, certain classes of descriptors might be usable for any sample, reducing the annotation cost a lot (as we can compute it from a single annotated sample and then reuse it for other samples). Knowing which descriptors are able to do that remains an open question, but the current submission shows that descriptor-based supervision is feasible, henceforth motivating that search for more powerful and generalizable descriptors.
>
> > The paper is very well written, but there are many grammar mistakes (e.g. been showed; capable to get) for which I would recommend careful proofreading.
>
> We thank the reviewer for pointing those out. We will perform several careful passes when the camera ready version gets finalized.
>
> > Comments in the abstract about biomarker discovery, interpretability and invariance across data are very interesting. Have the authors investigated to which extend generalization across imaging modalities apply especially when major covariate shifts are present? would the approach still require forms of domain adaptation?
>
> We have not investigated it yet, but it is something we want to study. More precisely, we wonder if some _patient-wise_ descriptors can be used across modalities, or if, for some tasks, some even higher-order descriptors could be valid for a broad range of scans.
>
> > Looking at the results reported in Table 1, performances on myocardium and prostate segmentation are substantially lower - it would be interesting to have a figure which shows how segmentation looks like in the best and worse cases.
>
> For the camera ready, we will add a complementary Figure in the appendix, with more visual examples and spanning more failure cases.
>
> > A follow up question would involve thoughts on how segmentation could be improved, would a larger training set be required, or additional shape descriptors? It is interesting that network initialization had more than an effect on training stability.
>
> We will first investigate, with the same annotation budget, even higher-order descriptors (as shape moments can go much higher), to see if some are fit to fill that gap.
>
> Another avenue, described in the top-level comment, might be to "revisit" existing datasets: mining shape descriptors from existing annotations might give valuable and usable supervision for unannotated examples (same patient, but different scan; or descriptors valid for multiple scans and patients).

---

### Author Response · Authors · 2021-03-18
**Top-level response**

We thank all reviewers for their constructive and kind reviews. Our response will be structured as follow:
- one top-level response, describing a direct application of the method, that we will perform in a future journal extension;
- one reply per reviewer.

### 3D extension: time independent shape descriptors
As mentioned in the paper initially, all shape descriptors can be extended (quite easily$^1$) to 3D. In the case of the ACDC dataset, this can be very powerful: the training data comes from actual 4D Cine-MRI scans (3D images over time), with annotations for two time points at the beginning of the systole and diastole (when the heart is at its biggest and smallest, respectively).

By computing the descriptors at those two extremes, one can get _patient-wise_ (and not _volume-wise_) upper and lower bounds for our descriptors, valid at any timestep. [_The following plots_](https://github.com/HKervadec/shape_descriptors/raw/master/patient001.png) (click to enlarge) show an ellipsoid per class (based on the _average distance to the centroid_, $\mathfrak D^{(k)}$, and centered around its centroid $\mathfrak C^{(k)}$). The thick lines represent the shift of the centroid between the two phases.

We can clearly see the 'shrink' between the two phases, and the slight shift of the centroids toward the center of the scan.

![patient001.png](https://github.com/HKervadec/shape_descriptors/raw/master/patient001.png)

The bounds given by those two annotations allow us to constraint the remaining unannotated 3D volumes at train time; increasing the training set size by one order of magnitude. (The initial, pixel-wise annotated subset could be supervised either with traditional pixel-wise losses, or with the shape descriptors that we propose.) The benefits are clear, especially in low-data settings (e.g., only 5 annotated patients out of the 70 that we used).

For a fair comparison, we will test some label-propagation methods (we have not decided which ones yet, as we are still in the bibliographic stage to select the most relevant methods) to tentatively annotate the remaining 3D volumes, and use cross-entropy on those propagated labels. However, any label-propagation method will necessarily suffer from the following limitations:
- sensitivity to the time-resolution--a longer 'gap' between 3D volumes means more movement, which makes the propagation harder and more error-prone;
- errors can accumulate over time, as we go from $t$ to $t+1$, $t+2$, ..., $t+n$.

On the contrary, the bounds for each shape descriptor, given by the systole and diastole annotations, will be valid regardless of the time resolution or the number of unannotated timesteps. Training in 3D can be performed as in [1], by regrouping all 2D slices from the same 3D volume in a single batch (with current hardware, one can fit  up to a `100x256x256` volume this way, if using an efficient network), which enables the easy computation of descriptors in 3D.

---
$^1$: Extending to 3D usually requires taking into account the spatial resolution, as it can vary a lot between slices (around 1mm between two pixels of the same slice, but 10mm between each slice, for instance). The other modifications (adding another axis) are usually trivial.

[1] Constrained-CNN losses for weakly supervised segmentation, Kervadec et al, MedIA 2019

---

### Decision · Program_Chairs · 2021-03-31

**Decision:**

Accept

**Comment:**

Congratulations your paper has been selected as a long oral.